# BusyBot: Learning to Interact, Reason, and Plan in a BusyBoard Environment

**Zeyi Liu**      **Zhenjia Xu**      **Shuran Song**
Columbia University, New York, NY, United States
https://busybot.cs.columbia.edu/

**Abstract:** We introduce BusyBoard, a toy-inspired robot learning environment that leverages a diverse set of articulated objects and inter-object functional relations to provide rich visual feedback for robot interactions. Based on this environment, we introduce a learning framework, BusyBot, which allows an agent to jointly acquire three fundamental capabilities (interaction, reasoning, and planning) in an integrated and self-supervised manner. With the rich sensory feedback provided by BusyBoard, BusyBot first learns a policy to efficiently interact with the environment; then with data collected using the policy, BusyBot reasons the inter-object functional relations through a causal discovery network; and finally by combining the learned interaction policy and relation reasoning skill, the agent is able to perform goal-conditioned manipulation tasks. We evaluate BusyBot in both simulated and real-world environments, and validate its generalizability to unseen objects and relations.

**Keywords:** Manipulation, Learning Environment, Reasoning

## 1   Introduction

Learning through physical interactions plays a critical role in human cognitive development [1, 2, 3]. For instance, a well-designed toy like the "busyboard" (Fig. 1a) can provide an effective learning environment for children to develop fundamental manipulation and reasoning skills: the rich and amplified sensory feedback encourages children to actively explore and interact; and the observed inter-object functional relations (e.g., a switch turns on a light) facilitate the development of reasoning and task solving skills.

In this paper, we aim to provide a similar learning environment for embodied artificial agents, the **BusyBoard** environment, where agents learn to discover the underlying relations of objects through informative interactions and plan for goal-conditioned tasks. While simple at the first glance, this relational environment provides an *integrated* tool for learning and evaluating three critical capabilities of an embodied intelligent system:

- **Interact:** The ability to infer action affordances from visual observations – knowing where and how to manipulate an object to effectively change its state. Learning this skill through visual feedback is particularly hard for small-displacement objects (e.g., switches), whose appearance changes can be subtle even under effective actions.

- **Reason:** The ability to reason about inter-object functional relations (e.g., pressing a button turns on a light). In particular, the agent should learn to infer the relations by observing and predicting future states of the environment, without using the ground-truth relations as supervision.

- **Plan:** The ability to use the learned manipulation and reasoning skills in goal-conditioned planning tasks, in other words, generating a sequence of actions to transform the environment from a random initial state to a given goal state.

To learn these skills from the environment, we propose the **BusyBot** framework that acquires the above three capabilities through **self-supervised** interactions. To acquire the manipulation skill, the algorithm learns a visual affordance model that infers effective action candidates through visual feedback. To reason about inter-object functional relations, the algorithm infers a functional scene

6th Conference on Robot Learning (CoRL 2022), Auckland, New Zealand.

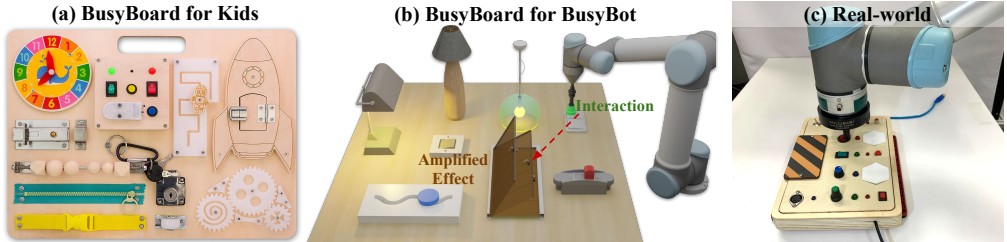

Figure 1: **BusyBoard Environments** inspired by toys for children, an integrated tool for learning and evaluating a robot's capabilities in interaction, reasoning, and planning.

graph and predicts the future states through a causal discovery network. Finally, to accomplish goal-conditioned manipulation tasks, the algorithm combines the learned action affordances, inter-object functional relationship, and dynamics to plan its actions with a model predictive control (MPC) framework. In summary, our contributions are two-fold:

- We introduce a new learning environment for embodied agents, **BusyBoard**, which features a diverse set of articulated objects with typical- and small- displacement joints, and rich inter-object functional relations.

- We propose **BusyBot**, an integrated learning framework which allows an embodied agent to acquire interaction, reasoning, and planning skills through self-supervised learning.

## 2    Related Work

**Simulation environments for robot learning.**  Simulation environments are crucial for advances in robot learning. However, most of the existing simulated environments are developed for specific tasks or capabilities, such as navigation [4, 5, 6, 7], manipulation [8, 9, 10, 11, 12], causal reasoning in 2D [13, 14], or high-level task planning [15]. Inspired by human toys, our BusyBoard is an integrated environment that is compact and relevant to real-world applications, where an embodied agent can jointly learn three critical capabilities: interaction, reasoning, and planning.

**Learning interaction policy.**  The ability to interact with a diverse set of objects is critical for many robotics tasks. Different methods have been proposed to learn interaction polices through human demonstrations [16, 17, 18] or self-guided explorations [19, 20, 21, 22]. However, most prior works have been ignoring a set of common but challenging objects: small-displacement objects. When interacting with these objects (e.g., switches), the effectiveness of an action often cannot be observed from the object's own visual appearance. In this work, we address this challenge by taking advantage of BusyBoard, which amplifies action effects through responder objects and enables learning by enriching the supervision signal.

**Inferring inter-object functional relations.**   Perceiving and understanding objects individually is often not sufficient for a lot of real-world applications that involve environments with multiple objects, and understanding inter-object relations [23, 24, 25] is a crucial skill for efficient planning. In this work, we will tackle the problem of uncovering the inter-object functional relationship, as defined by Li et al. [26]. One common approach is to induce changes through interventions and iteratively construct a functional scene graph [27]. More recently, Graph Neural Networks (GNNs) have been demonstrated to be promising for extracting the underlying structural causal model (SCM) [28, 29, 30, 31, 32] and predicting future dynamics from motions [25]. In our work, we further demonstrate that GNNs are able to infer inter-object functional relations from changes in visual appearances. We also show that the inferred inter-object functional relationship and scene dynamics can assist action planning for downstream goal-conditioned manipulation tasks.

**Hierarchical Planning in Dynamic Environment.**  Hierarchical planning has been proven effective for long-horizon planning in dynamic environment by decomposing a plan into several high-level sub-goals and low-level actions [33, 34, 35]. Some prior works have proposed using a high-level logic-based planner with learned planning operators (POs) to encode preconditions, actions, and effects [36, 37] in a RL setup. However, both works assume abstract action and state space (e.g., grid world). In contrast, our work learns physical actions to manipulate articulated objects from image observations, making our work easier to be deployed in the real world. In addition, prior works model inter-object relations implicitly from state transitions, while we explicitly learn a scene graph which helps better generalization to new tasks and environments.

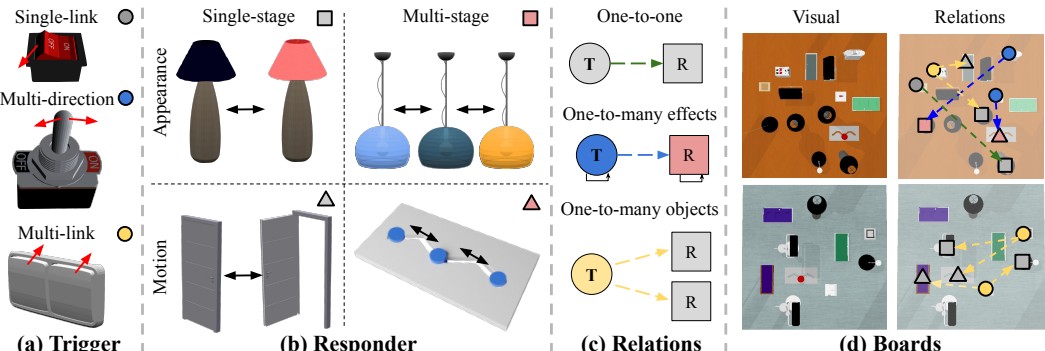

Figure 2: **BusyBoard Environment** is procedurally generated using articulated objects (a,b) with randomly sampled inter-object functional relation pairs between trigger and responder objects (c). (d) shows example boards and the underlying functional relations.

# 3 The BusyBoard Environment

**Trigger and responder objects.** As shown in Fig. 2, the BusyBoard is procedurally generated using object URDF models. For trigger objects, we select switch instances from the Partnet-Mobility dataset [12], including small-displacement instances (with small displacements upon interaction), multi-direction instances (contain one movable link that can be pushed to multiple directions), and multi-link instances (contain multiple movable links). We use other object categories (e.g., lamp, door, tracktoy) as responder objects, which can be either single-stage or multi-stage. A single-stage responder has only two possible states defined by either appearance (e.g., lamp on or off) or joint state (e.g., door open or closed). A multi-stage responder has multiple possible states (e.g., multiple light colors of a lamp responder or multiple joint positions of a tracktoy responder).

**Relations.** We introduce three types of inter-object functional relations:

- **One-to-one:** one trigger controls one single-stage responder.
- **One-to-many effects:** one trigger controls multiple effects on one responder. The trigger is a multi-direction object and the responder is a multi-stage object.
- **One-to-many objects:** one trigger controls multiple responders. The trigger is a multi-link object (a switch with multiple buttons), and each link controls one single-stage responder.

The inter-object functional relations enable an important property of BusyBoard: in addition to providing visual feedback on the interacted trigger object (e.g., appearance change on a button after being pressed), BusyBoard also amplifies the effect of an action with responder objects (e.g., a light turns on after the button is pressed). This is especially useful for learning manipulation policies for objects with small displacements upon interaction, for which state changes are often hard to observe.

Note that in goal-conditioned tasks, for both "one-to-many effects" and "one-to-many objects" relations, the algorithm not only needs to know the trigger object to interact with, but also the direction and position of the action to execute. In this paper, we use "one-to-many" to refer to the super-set of both categories. We exclude many-to-one relations to eliminate possible ambiguities.

# 4 The BusyBot Framework

The goal of BusyBot is to meaningfully interact with the BusyBoard environment (§4.1), infer the inter-object functional relations and dynamics through these interactions (§4.2), and eventually perform goal-conditioned manipulation using the learned interaction policy, relations, and dynamics. (§4.3). We will discuss each module in detail below.

## 4.1 Interact: Learning to Interact with Amplified Effects

The goal of the interaction policy $\pi$ is to take a top-down depth image $o_t \in \mathbb{R}^{W \times H}$ as input and generate an action $a_t$ at each step $t$: $\pi(o_t) \to a_t$. The action is parameterized by an end effector (i.e., a suction-based gripper) position and a moving direction $a_t = (a_t^{\text{pos}}, a_t^{\text{dir}})$, where $a_t^{\text{pos}} \in \mathbb{R}^3$ is the 3D coordinate and $a_t^{\text{dir}} \in \mathbb{R}^3, (||a_t^{\text{dir}}|| = 1)$ is a unit vector in 3D indicating the moving direction of the end effector. The moving distance is incrementally assigned until reaching a pre-defined limit. The

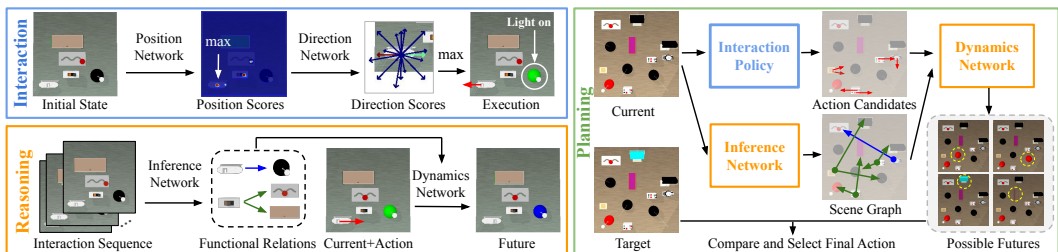

Figure 3: **BusyBot Overview.** [Interaction] infers a sequence of actions from visual input to efficiently interact with a given scene. [Reasoning] infers a functional scene graph (i.e., inference network) and predicts future states (i.e., dynamics network). [Planning] uses the trained manipulation policy network (learned from multiple boards), relation inference and dynamics network (extracted from the specific board) to plan actions for reaching the target state.

policy is considered effective if it can 1) successfully trigger changes in responder objects, and 2) interact with different objects to explore novel states of the board.

**Interaction policy.** The interaction policy is modeled by two neural networks: a position network and a direction network, which follows a similar formulation as UMP-Net [20] to jointly learn the action position and direction. The position network takes a depth image as input and outputs per-pixel position affordance $P \in [0,1]^{W \times H}$, which indicates the likelihood of an effective interact position. The direction network takes in the depth image and the selected action position (represented as a 2-D Gaussian distribution centered around the corresponding pixel location of the 3-D action position) and outputs a score for each direction candidate $r(a_t^{\text{dir}}) \in [0,1]$. We uniformly sample 18 directions in SO(3) as direction candidates. Since it is often hard to identify the state of small-displacement objects from visual observations, the agent executes both the selected direction and its opposite direction.

**Supervision.** Unlike UMP-Net [20] that requires object state from simulation to compute reward, BusyBot uses a simple self-supervised reward computed from image difference, which is enabled by the amplified effects in the BusyBoard environments:

$$r_{\text{img}}(a_t^{\text{dir}}) = \begin{cases} 1 & \text{if } \sum_{i=1}^{H} \sum_{j=1}^{W} I(o_{ij}, o'_{ij}) > \delta \\ 0 & \text{if } \sum_{i=1}^{H} \sum_{j=1}^{W} I(o_{ij}, o'_{ij}) \leq \delta \end{cases} \quad I(o_{ij}, o'_{ij}) = \begin{cases} 1 & \text{if } o_{ij} = o'_{ij} \\ 0 & \text{if } o_{ij} \neq o'_{ij} \end{cases} \quad (1)$$

where $o$ and $o'$ denote RGB image observations before and after the action execution. $\delta$ is a threshold specifies the minimum number of different pixels. We use binary cross-entropy (BCE) loss between the inferred action score $r(a_t^{\text{dir}}) \in [0,1]$ and the ground-truth reward computed from image observations $r_{\text{img}}(a_t^{\text{dir}}) \in [0,1]$.

**Exploration.** At the early stage of training the position and direction inference network, we use the epsilon-greedy method to encourage random exploration. Additionally, in order to prevent the model from only selecting the position that has the highest affordance score, we apply the Upper Confidence Bound (UCB) Bandit algorithm on the inferred position affordance. Given the per-pixel position affordance score $P(i,j)$, the updated score is $P'(i,j) = P(i,j) + c\sqrt{\frac{ln(t)}{N}}$, where $c = 0.5$, $t$ is the number of past steps, and $N$ is the numeber of times when the pixel $(i,j)$ falls in the $M \times M (M = 10)$ window centered around each previously selected pixels.

### 4.2 Reason: Learning to Discover Inter-object Relations by Predicting the Future

The reasoning module takes in RGB image sequences of the agent's interactions (§4.1), infers the inter-object relations, and predicts future dynamics, which would guide goal-conditioned planning (§4.3). To accomplish this goal, we adopt and modify the V-CDN model [25].

The inference network is implemented with three Graph Neural Networks (GNNs) to extract functional relations as a scene graph. Each object $O_i$ corresponds to a node $i$ in the graph, with a node input $n_i^{1:T}$, where $T$ is the first $T$ frames in an interaction sequence. Unlike the V-CDN model [25] that uses keypoint locations as node features, BusyBot uses both objects' visual features and locations, which allows the network to represent both motion change (e.g., door opens) and appearance change (e.g., light turns on) of the responders. The first GNN learns spatial node and edge embeddings at each step, which are concatenated with 256-dimensional embeddings of the executed actions $a_i^{1:T}$ output from an MLP layer. The combined embeddings are then aggregated over temporal dimension using a

1-D convolution network and input to the second GNN which predicts a probabilistic distribution over edge types $e^d = \{e^d_{ij}|e^d_{ij} \in \mathbb{R}^2\}^N_{i,j=1}$ (index 0 indicates no relation and index 1 indicates has relation). Conditioned on the edge types, the third GNN predicts 32-dimensional edge embeddings $e^h = \{e^h_{ij}|e^h_{ij} \in \mathbb{R}^{32}\}^N_{i,j=1}$ which store history dynamics associated with each edge.

The dynamics network is a Graph Recurrent Network (GRN) that predicts the next state $n^{t+1}$ given the current observation $n^t$, the executed action $a^t$, and the edges $E = \{e^d, e^h\}$ from the inferred functional scene graph. The inference and dynamics network are jointly trained under the objective to minimize the mean squared error (MSE) between predicted and ground-truth object features. $f^I_\phi$ denotes the inference model parameterized by $\phi$, $f^D_\psi$ denotes the dynamics model parameterized by $\psi$.

$$L = \min_{\phi,\psi} \sum_t MSE(n^{t+1}, f^D_\psi(n^t, a^t, f^I_\phi(n^{1:T}, a^{1:T}))) \qquad (t \geq T) \qquad (2)$$

**Data collection.** The interaction dataset used in the reasoning module is collected the using learned interaction policy: at each step, positions with affordance scores above a threshold are grouped into clusters using the K-means clustering algorithm (with $k$ being the maximum possible number of movable links on all busyboards), and positions with the highest score in each cluster are selected as position candidates. Conditioned on each position candidate, direction with the highest affordance score will be selected to form the final action candidates, from which a candidate will be randomly chosen to execute. For each board environment, RGB image observations of 30 interaction steps are generated. In addition, to prevent the model from overfitting on board appearances, we ensure that every 20 board environments share the same initial visual appearance but with different inter-object functional relations.

### 4.3 Plan: Goal-conditioned Manipulation with Relation Predictive Agent

Given an initial and target image of a BusyBoard, the task is to have BusyBot infer 1) which object(s) to manipulate; 2) what action(s) to execute in order to successfully reach the target state.

Using the data collection method as discussed in the reasoning module, the agent infers the action candidates and generate an interaction sequence of 30 images, which is input to the inference network to obtain the functional scene graph. Then we consider three options to plan for goal-conditioned tasks: 1) **Relation agent**, at each step, identify a responder that needs to be changed, and find the corresponding trigger based on the functional scene graph. This method is similar to the idea of Li et al. [26]. However, the agent might have trouble handling one-to-many relations. To solve this issue, we propose 2) **Predictive agent** that uses the dynamics network from the reasoning module and choose the action that minimizes the L2 distance of the predicted next state and the target state. However, the predictive agent may have difficulty generalizing to novel object instances, due to the difficulty of predicting unseen dynamics. 3) Unlike piror works that only use either predicted dynamics or functional scene graph for planning, our final method **BusyBot** combines the relation and predictive agent, where action candidates are first filtered based on the functional scene graph and then selected based on dynamic predictions. More discussions are provided in Sec. 5.

## 5 Evaluation

We evaluate BusyBot with both simulated (Fig. 4) and real-world busyboards (Fig. 6). In simulation experiments, we set up the following environments: **a) Training Board**: for training interaction and reasoning module. **b) Novel Config**: testing board with training object instances but in new configurations, which include new inter-object functional relations, position and orientation of objects, board color and texture; **c) Novel Object**: both object instances and board configurations are novel. In total, we generate 10,000 training boards, 2,000 boards with novel configurations, and 2,000 boards with novel object instances. We generate 30 interaction images for each board, where 23 images are reserved for relation inference and the rest for future predictions. As for object instances, we use 41 switches, 10 doors, 5 lamps, and 2 tracktoy objects, split into training / testing with ratio: 32/9, 5/5, 3/2, 1/1. The setup for real-world evaluation is described in Sec 5.4

### 5.1 Interaction Module Evaluation

To evaluate the interaction policy network, we compute the average precision and recall of the inferred actions for the boards, where precision = # successful actions / # total proposed actions, and recall = # successfully interacted objects / # total interactable objects. We compare the following methods:

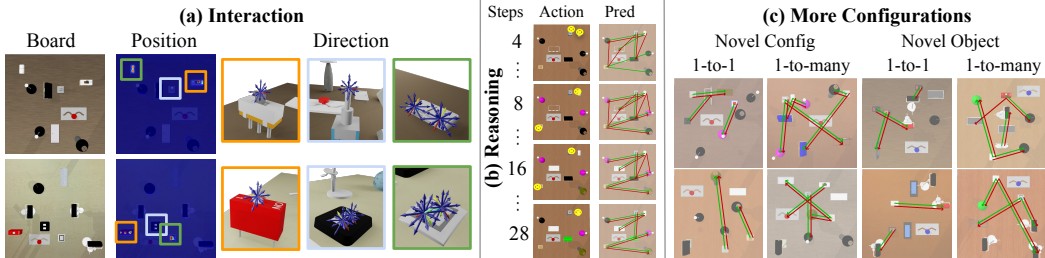

Figure 4: **Qualitative Results**, (a) action affordances (b) interaction steps and corresponding reasoning results (c) More reasoning results →: inferred inter-object functional relations. →: ground truth.

- **Oracle (joint state supervision):** Interaction policy supervised on joint states. This is considered as the oracle because changes in joint states are directly obtained from simulation.
- **RGB:** A baseline that takes RGB images instead of depth images as input to the model.
- **w/o responder:** Interaction policy supervised on visual feedback but no responder effects.
- **w/o exploration:** An ablated version of BusyBot without using UCB for exploration.

**Results and Analysis.** From Tab. 1, we can see that [w/o responder] achieves poor performance since visual feedback of small-displacement objects alone is insufficient for learning. In contrast, [BusyBot] is able to achieve comparable performance as the oracle, which validates our hypothesis that triggered responder effects can assist the model in learning good interaction policy by amplifying the visual feedback. We also demonstrate the effectiveness of our exploration method by showing the recall of [w/o exploration] drops by more than 45% than that of [BusyBot]. From the [RGB] baseline, we find that color observations alone struggle to learn interaction policy for instances such as switches that are all-white. Moreover, the performance of [RGB] drops significantly for novel objects, implying that the model trained with RGB observations overfits on colors and lacks generalizability.

|  | Novel Config | | Novel Object | |
|---|---|---|---|---|
|  | Prec | Recall | Prec | Recall |
| Oracle | 91.8 | 82.4 | 79.5 | 90.2 |
| RGB | 62.3 | 63.1 | 9.50 | 17.0 |
| w/o responder | 0.71 | 0.65 | 0.24 | 0.49 |
| w/o exploration | **94.2** | 33.7 | 81.9 | 38.6 |
| BusyBot | 90.1 | **80.1** | **82.6** | **84.8** |

Table 1: **Performance of Interaction Policy**

## 5.2 Relation Reasoning Module Evaluation

The reasoning module is evaluated by the following metrics: 1) Relation inference accuracy, measured by the precision (**Edge-P**) and recall (**Edge-R**) of the inferred functional relation pairs. 2) Future state prediction accuracy (**Pred-A**), measured by the percentage of correct future state predictions. We compare the following alternatives:

- **w/o inference:** An ablated version of the model without the inference network. The dynamics network takes in all history interaction data and directly predicts the next state.
- **w/o exploration:** An ablated version of our method, where the input of the reasoning network are data collected under an inferior interaction policy.

|  | Training Board | | | Novel Config | | | Novel Object | | |
|---|---|---|---|---|---|---|---|---|---|
|  | Edge-P | Edge-R | Pred-A | Edge-P | Edge-R | Pred-A | Edge-P | Edge-R | Pred-A |
| w/o inference | - | - | 79.2 | - | - | 36.2 | - | - | 7.04 |
| w/o exploration | 55.6 | 51.0 | **89.6** | 74.6 | 3.10 | 14.3 | 75.1 | 0.96 | 11.6 |
| BusyBot | **95.8** | **100** | 88.1 | **95.5** | **99.7** | **73.8** | **85.0** | **99.5** | **31.0** |

Table 2: **Performance of Reasoning Module.** For BusyBot, while the future state prediction accuracy (Pred-A) decreases for unseen board appearances (novel config, novel object), the reasoning module is still able to reliably infer the inter-object functional relations (Edge-P, Edge-R) in novel scenarios.

**Results and Analysis.** Compared to [w/o inference], we see that without inferring the inter-object relations, the model overfits on the training data and generalizes poorly to novel boards. We also observe that with bad interactions [w/o exploration], the reasoning model is not able to uncover the relations accurately and make correct future state predictions. In comparison, our model generalizes well to novel board configurations and achieves performance comparable to that of the training board. This demonstrates that a good interaction policy helps the agent uncover the correct inter-object

functional relations, which then helps the agent to understand scene dynamics. For boards with novel object instances, even though the future state prediction accuracy drops by around 40% than the seen instances (which is expected since the object features are never seen by the dynamics model), the relation inference accuracy is still comparable. The performance on novel boards verifies that the model's ability to infer inter-object functional relationship can transfer to new scenes and objects.

## 5.3 Goal-conditioned Manipulation

We generate 50 one-to-one tasks and 50 one-to-many tasks for each type of board (training, novel config, novel object). One-to-one tasks contain only two-state triggers, and thus only require the algorithm to identify the correct trigger (similar task studied in IFRexplore [26]). One-to-many tasks contain both multi-direction and multi-link triggers that require the agent to not only identify the correct trigger, but also infer the correct action to manipulate the trigger (e.g., the correct button position or pushing direction).

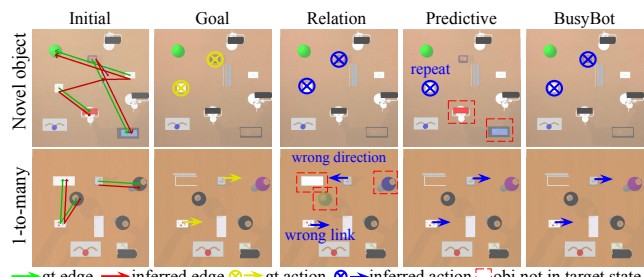

→ gt edge  → inferred edge  ⊗ gt action  ⊗→ inferred action  ⊡ obj not in target state

Figure 5: **Goal-conditioned Manipulation.** Compared to the predictive agent, the relation agent generalizes better on novel objects, while struggles in handling one-to-many relations. Our method BusyBot combines the advantages of both agents.

**Metrics & Baselines.** We measure object-level success rate on both one-to-one and one-to-many manipulation tasks for each type of board. The success rate is defined at object level at the end of an interaction sequence with a maximum of 8 steps. Success rate = # affectable responders in goal state / # total affectable responders. We compare the three agents as discussed in 4.3, along with two learning-based agents using behavior cloning (BC) and proximal policy optimization (PPO). More details can be found in supp.

|  | Training | | Novel Config | | Novel Object | |
|---|---|---|---|---|---|---|
|  | 1-to-1 | 1-to-m | 1-to-1 | 1-to-m | 1-to-1 | 1-to-m |
| PPO | 91.2 | 88.5 | 62.3 | 54.3 | 59.9 | 55.7 |
| BC | 95.4 | **94.9** | 57.0 | 51.8 | 63.8 | 54.8 |
| Relation | 98.3 | 61.1 | 93.7 | 60.0 | 92.0 | 62.8 |
| Predictive | 97.7 | 67.5 | 91.0 | 67.0 | 89.0 | 58.2 |
| BusyBot | **98.3** | 71.0 | **93.7** | **69.4** | **92.3** | **64.9** |

Table 3: **Goal-conditioned Manipulation Result**

**Results and Analysis.** All agents achieve good performances on one-to-one tasks. This means that both the relation and dynamics learned by the reasoning module can generalize to novel board configurations and objects. The [predictive] agent achieves better performance on one-to-many tasks with seen object instances by leveraging future predictions to select the correct action to apply on the trigger object. In contrast, the [relation] agent can only identify the trigger object but not the exact action (e.g., which link to interact with or which direction to push). On the other hand, the [relation] agent performs slightly better than the [predictive] agent on all one-to-one tasks and boards with novel objects, when the dynamics model sometimes fails to predict the correct next state. This shows that inter-object functional relationship can generalize to scenarios when future predictions are not reliable enough to assist planning. The result of [PPO] and [BC] indicate that RL-based agents fail to generalize to boards with novel configurations or objects, and it is thus critical to learn an explicit representation of inter-object functional relations.

## 5.4 Real-world Experiments

**Setup.** We test the trained model on a busyboard in real world with robot interactions (Fig. 6). The board consists of 3 trigger objects (switches) and 3 responder objects (LEDs). Objects outside the effective region are ignored. We manually modified the underlying inter-object functional relations of the board by rewiring the objects. We test with 6 different configurations including 4 one-to-one and 2 one-to-many configurations. For each configuration, the robot interacts with the board for 30 steps and the rollout is grouped into 6 overlapping and continual sub-sequences, each of which has a length of 25. In total, we generate a real-world dataset of 36 sequences with 108 inter-object functional relation pairs for evaluating the reasoning module.

**Results.** Fig. 6 (c) shows that the algorithm is able to refine inter-object functional relations (reducing additional edges) through interactions. The precision and recall of inferred relations are

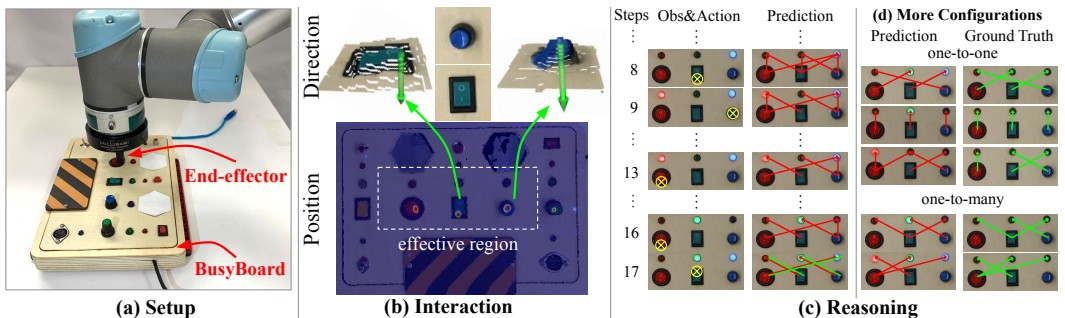

**(a) Setup**     **(b) Interaction**     **(c) Reasoning**

Figure 6: **Real-world Busyboard.** We test the trained models on a real-world busyboard with robot interactions (a), and show that our algorithm is able to discover the inter-object relations (c) through interactions (b).

**93.9%** and **100%**, respectively. All inter-object functional relations can be discovered by our model, with only a few additional pairs predicted. The result shows that the relation reasoning ability of the model is transferable to real-world scenarios. More results can be found in supp.

## 5.5 Application in Simulation Home Environments

To demonstrate the learned skills can be applied beyond the BusyBoard environment, we further test our reasoning model in 2 kitchen scenes from AI2THOR [15]: a) a stove with multiple controls and b) a room with multiple objects. Following the same evaluation protocol, we let the agent interact with the environments for a few steps and use the trained reasoning model to infer the functional scene graphs. We observe that the

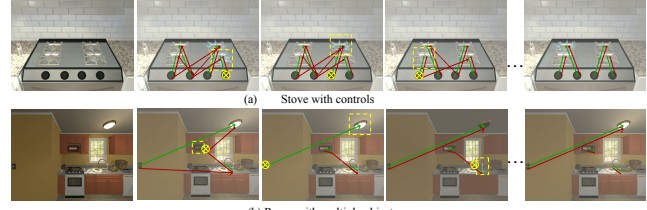

Figure 7: **Application in AI2THOR Home Environments.** The figure shows the interactions (in yellow), corresponding state changes, ground truth inter-object functional relationships (in green), and inferred relationships (in red).

algorithm achieves similar performance to boards with novel objects (shown in 5.2): while the algorithm cannot perfectly predict the future states of the object (due to out-of-distribution object visual appearance), it is able to infer the correct edges through interactions, without the need of fine-tuning. The result demonstrates the generalization ability of our proposed environment and algorithm to new domains and applications.

## 5.6 Limitation and Future Work

While the BusyBoard environment is inspired by toys, it still lacks certain diversity and complexity in real-world toys. For example, real-world toys are often designed with multi-sensory feedback such as sound and tactile, while our environment focuses on visual effects only. Several assumptions made by BusyBot could also be relaxed for more general applications. First, BusyBot assumes full observability of objects in a single image. Future work may consider using a 3D scene representation that integrates multi-view observations to handle larger-scale environments. In addition, the interaction module assumes trigger objects can reach all states through single-step actions. Future work could consider learning a more general manipulation policy [20] to accomodate objects that require a sequence of actions to manipulate. Finally, the relation reasoning module assumes objects are detected. Though the assumption is valid in our setup, it may not hold for cluttered scenes.

## 6 Conclusion

We propose a toy-inspired relational environment, BusyBoard, and a learning framework, BusyBot, for embodied AI agents to acquire interaction, reasoning, and planning abilities. Our experiments demonstrate that the rich sensory feedback in BusyBoard helps the agent learn a policy to efficiently interact with the environment; using the data collected under this interaction policy, inter-object functional relations can be inferred through predicting future states; and by combining the ability to interact and reason, the agent is able to perform goal-conditioned manipulation tasks. We verify the effectiveness and generalizability of our method in both simulation and real-world setups.

**Acknowledgments:** The authors would like to thank Huy Ha, Cheng Chi, Samir Gadre, Neil Nie, and Zhanpeng He for their valuable feedback and support. This work was supported in part by National Science Foundation under 2143601, 2037101, and 2132519, and Microsoft Faculty Fellowship. We would like to thank Google for the UR5 robot hardware. The views and conclusions contained herein are those of the authors and should not be interpreted as necessarily representing the official policies, either expressed or implied, of the sponsors.

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
