# OpenReview forum: "BusyBot: Learning to Interact, Reason, and Plan in a BusyBoard Environment"
_robot-learning.org/CoRL/2022/Conference — CoRL 2022 Poster_

### Official Review · Reviewer_55uE · 2022-07-22

**Originality:** Good
**Technical Quality:** Good
**Clarity Of Presentation:** Very Good
**Impact:** 2

**Recommendation:**

Weak Accept: I recommend accepting the paper, but will not argue for my recommendation if the majority of other reviewers have a different opinion.

**Summary:**

This work proposes a framework for generating simple learning environments both in simulation and in the real world, dubbed BusyBoard. BusyBoard consists of sets of switches and objects, some of which are articulated, that can be interacted with and uses lights to amplify the effects of many of the small actions (e.g. pushing switches) making them more observable. BusyBoard instances are generated procedurally from URDF files making the environments amenable to many variations on the BusyBoard theme.

The authors also propose a baseline for the BusyBoard environment dubbed BusyBot. BusyBot learns action affordances from depth, learns inter-object interactions, and then creates and executes plans in the BusyBoard enviornment.

**Issues:**

- Consider adding comparisons to existing agent-learning approaches for BusyBot. As is, it's difficult to evaluate the value of BusyBot and most of the paper's contribution centers around the BusyBoard domain.

**Quality Of The Limitations Section:**

Limitations are addressed clearly

**Reviewer Expertise:**

4: The reviewer is confident but not absolutely certain that the evaluation is correct

**Robotics Focus:**

Sufficient demonstration on hardware

**Strengths And Weaknesses:**

Strengths
- The BusyBoard framework complements typical toy domains, such as gridworlds and block stacking, by providing a relatively rich set of inter-object interactions in a small and feasibly explorable space with relatively dense reward.
- BusyBoard's procedural nature lends itself nicely to explorations of transfer learning and distribution shift.
- BusyBot appears to be a strong baseline for this domain (with one of the biggest assumptions being the sufficiency of single-step actions to realize meaningful change in the world).
- The paper contains an interesting hardware implementation of the both the BusyBoard domain and BusyBot.

Weaknesses
- By design the BusyBoard domain does not attempt to expose challenges such as partial-observability and sparse long-horizon rewards. While the authors are explicit about this design choice, these are two extremally fundamental difficulties in robotics.
- The BusyBot baseline is not well explored in the evaluation. While the design appears reasonable, virtually no comparison was made with existing learning methods that may be applicable to this task. As a result, this paper's primary contribution centers on BusyBoard; it's hard for a reader to judge the effectiveness of BusyBot.

**Summary Of Recommendation:**

I advocate for acceptance of this work. I think the BusyBoard domain, while simple, has some value to the community as a platform for exploring learning in constrained settings. That being said, BusyBoard, by design, hides much of the complexity present in less structured environments. As a result, progress in the BusyBoard domain may not be a particularly good proxy for progress in real-world tasks.

---

> ### Author Response · Authors · 2022-08-24
> **Author's Response to Reviewer 55uE**
>
> Thank you for the thoughtful feedback. We are glad to see the value of the BusyBoard environment is recognized! Here are the answers to your questions and reference to the updated sections in the paper.
>
> ### **The BusyBot baseline is not well explored in the evaluation.**
>
> Based on the reviewers’ suggestion we have included additional baselines including:
> 1. a behavior-cloning agent (i.e., BC agent) for goal-conditioned manipulation (see Table 3) with oracle demonstrations.
> 2. a reinforcement learning agent (i.e., a PPO agent) for goal-conditioned manipulation (see Table 3).
> The result shows that while the BC agent and RL agent are able to overfit on training boards when optimizing success rate as a reward, they fail to generalize to on boards with novel configurations or novel objects. This result validates the importance of learning explicit representation of inter-object relations.
> 3. an RGB observation-only baseline for manipulation (see Table 1), validating the choice of using depth input to train the interaction model, which captures structure information and enables better generalization.
> Our goal in designing BusyBot is to provide a solid baseline for future works to compare and study the task. We hope these additional baselines could provide a complete picture of the BusyBot algorithm and inspire future works.
> ### **Partial-observability and sparse long-horizon rewards.**
>
> Thank you for the feedback, we have added further discussion in the paper to reflect this limitation.

---

> > ### Comment · Reviewer_55uE · 2022-08-26
> > **Response to Author's comments**
> >
> > Thank you for your response, the additional baselines significantly improve the context in which BusyBot is presented.

---

### Official Review · Reviewer_9A6a · 2022-07-25

**Originality:** Fair
**Technical Quality:** Good
**Clarity Of Presentation:** Very Good
**Impact:** 3

**Recommendation:**

Weak Accept: I recommend accepting the paper, but will not argue for my recommendation if the majority of other reviewers have a different opinion.

**Summary:**

There are 2 main contributions of the paper.
* The first is the BusyBoard environment which allows generations of many tasks. The environment places articulated objects and creates inter-object functional relationships between the objects and responder objets.
* The second is a self-supervised learning framework called BusyBot that proposes to solve goal conditioned tasks in the above environment. The framework proposes a interaction policy using supervised learning, a reason module which collects interactions with the environment to create a scene graph using GNNs and a future dynamics model, and a planning agent which uses the above modules to perform goal conditioned tasks in the environment.


**Issues:**

* Clarify the issue of many-to-one tasks - were these tasks attempted or could the environment be modified to generate these sorts of tasks? Would the proposed method address these tasks?
* Could the authors clarify the choice of observation space. Why was only topdown depth chosen?
* It's hypothesized that the relation agent might have trouble handling 1-to-m relations (L196). Why is the 1-to-m result higher for novel objects for the relation agent than the predictive agent?
* Could the authors clarify how BusyBot could be useful outside of the 1-to-1 or 1-to-m scenarios presented in the env?
* Addressing or explaining limitations mentioned above in limitations section.

**Quality Of The Limitations Section:**

Additional details required

**Reviewer Expertise:**

3: The reviewer is fairly confident that the evaluation is correct

**Robotics Focus:**

Sufficient demonstration on hardware

**Strengths And Weaknesses:**

* The environment is interesting because it allows exercising integrated skills for embodied agents. Interaction, reasoning, and planning fused into one environment.
* The simplicity of the observations and action spaces (topdown depth and SE(3) pose) could be seen as a plus, to focus on the reasoning and planning parts of the problem.
* The ability of the environment to generate new task configurations to test novel placements and novel objects.
* The proposed BusyBot framework has a high success rate in sim and real and beats a few baselines proposed by the authors.
* The solution for each reasoning/planning/interaction components seems well thought out.
* Real world experiments.
* The paper overall is well written and clear.


Weaknesses
* The environment does not contain many-to-one relationships, which would be a very interesting setup to test for embodied agents, requiring agents with memory and potentially longer sequences and interactions.
* The observation is a top-down depth image. This modality may not be available in all cases for real world robotic systems. There are no ablations on this modality. It would be nice to see how the system compares with rgb observations for example, or a perspective view of the board.
* No vanilla RL or BC baselines. It would be interesting to see the performance of even a simple BC sequence model, trained on a sim oracle or human dataset. It could help answer the question: how necessary are the separation of interaction/reasoning/planning models?
* BusyBot is not evaluated on environments other than BusyBoard, so it’s unclear how generally useful this system could be. Is it “overfit” to this particular setup? A few details in particular that could be a concern for it’s usefulness:
* As mentioned above, many-to-one interactions are not included and could be common in real world tasks. Unclear if the method would work for this case.
* A few parts of the system, such as the k-means clustering hyperparam, could require tuning in different environment setups.
* Baselines are not tested on the real world.


**Summary Of Recommendation:**

I think the paper is borderline and I would consider changing my recommendation with a few clarifications and/or results. The environment and proposed method are solid and the experimental results support the authors conclusions overall. The environment is a good contribution, as a single environment that evaluates several skills in embodied agents.
The BusyBot results could be made more convincing. I think testing the proposed method on more complex many-to-one tasks or with more realistic modalities, or against a more diverse set of baselines would be more convincing.
I recommend addressing some of the weaknesses above to improve the paper.

---

> ### Author Response · Authors · 2022-08-24
> **Author's Response to Reviewer 9A6a**
>
> We thank the reviewer for detailed suggestions and feedback. We have provided additional experiments in the revised paper and we believe these additional baselines really add value to the paper by providing a complete picture of the task and baselines for future works.
>
> ### **Many-to-one relationships**
> We have included an additional evaluation on environments with many-to-one relations and discussed the potential ambiguity of such tasks. See Sec. 7 in the updated pdf.
> Our original board design only contains one-to-one and one-to-many relations and deliberately excludes many-to-one relations because many-to-one relations need more specific definitions. For n triggers to control 1 responder, we need to define $2^n$ states for the responder given the states of the triggers, and this is hard to sample in practice. If we loosely define many-to-one relations as multiple triggers that could control 1 responder, and whichever trigger manipulated later controls the state of the responder, we can easily incorporate this particular kind of “many-to-one'' relations into our environment.
>
> ### **Ablation using only RGB observations**
>
> Thank you for the suggestion! We have included this additional baseline in our paper (Table 1). Here we briefly summarize the result. First, using only RGB observation is not sufficient for some objects (switches that are completely white). Moreover, colors would also introduce bias and hurt generalization – the performance of the RGB baseline is significantly lower on unseen objects whose colors might be different from the training instances, while our depth-only method does not have this issue.
>
> ### **No vanilla RL or BC baselines.**
>
> Thank you for the suggestion! We have included BC and RL baselines in the updated paper (Table 3).  The result shows that the BC and PPO agents are able to overfit on training boards and achieve good results (better than BusyBot), but they fail to generalize to boards with novel configurations or novel objects. This experiment indicates that the explicit representation of inter-object relations is important for better generalization ability.
>
> ### **BusyBot is not evaluated on environments other than BusyBoard.**
>
> To demonstrate the learned reasoning skill can be applied beyond the scope of the BusyBoard environment, we further test our reasoning model in two kitchen scenes from AI2THOR with realistic home appliances and inter-object functional relations (adapted from prior work IFR-EXPLORE [1]). See updated pdf Sec. 8. In short, we found the reasoning module of BusyBot is able to infer the correct inter-object functional relations with no additional training on the two scenes.
>
> ### **A few parts of the system, such as the k-means clustering hyper-parameter, could require tuning in different environment setups.**
>
> In our experiment, we found that one set of parameters could work for a variety of environments with different number of objects. For example, we only need to set k in the k-means algorithm to the maximum possible triggers (movable links) over all scenes.
>
> ---
> [1] Li, Qi, et al. "IFR-Explore: Learning Inter-object Functional Relationships in 3D Indoor Scenes." arXiv preprint arXiv:2112.05298 (2021).

---

> > ### Comment · Reviewer_9A6a · 2022-08-26
> > **Response to Author's comments**
> >
> > Thanks for the clarifications and additional experiments. Based on the new info I will update my recommendation to a Weak Accept.

---

### Official Review · Reviewer_yhwf · 2022-07-31

**Originality:** Fair
**Technical Quality:** Good
**Clarity Of Presentation:** Good
**Impact:** 3

**Recommendation:**

Weak Reject: I recommend rejecting the paper, but will not argue for my recommendation if the majority of other reviewers have a different opinion.

**Summary:**

Self-contained busyboard environment with articulated objects and functional relations.
Self-supervised strategy: 1- learn policy to interact with environment, 2- collect data with policy and learn inter-object relations with causal discovery model 3- combine policy and causal model to learn goal-conditioned manipulation model.
motivation: busyboards amplify sensory feedback and have causal relations and so are ideal self-learning env for toddlers and robots.
8 sub-models: position network, direction network, 3 GNNs, 1 GRN, 2 agents, trained with thresholded image difference reward, MSE, and L2. Trained on 10k boards, eval’d on novel boards (2k) and objects (2k).
Results: interaction module: on par with “oracle”, benefits from responders vision and exploration. Relation reasoning: high accuracy even for novel configs and objects. goal-conditioned manipulation: good for 1-1, less for 1-many. real: high scores for 3 triggers and 3 lights.


**Issues:**

- minor: Lacking references for GNN papers?
- more articulate vision on how the proposed approach is applicable beyond this setup.
- code and sim available?


**Quality Of The Limitations Section:**

Additional details required

**Reviewer Expertise:**

4: The reviewer is confident but not absolutely certain that the evaluation is correct

**Robotics Focus:**

Sufficient demonstration on hardware

**Strengths And Weaknesses:**

strengths:
- A self-contained diverse environment for learning interactions, reasoning and planning is good for robotics research.
- publicly available code and simulation is nice (have not seen it available yet).
- The proposed solution seems to yield good results.

weaknesses:
- thresholded image difference as reward function: how would this be deployed in the real world?
- The learning method seems overly engineered and tuned for this specific problem, it seems unlikely that it would be useful for broad learning in the real world as is, a more unified model (as opposed to the 8 modules used here) or more end-to-end design. I understand that some engineering is ok for the sake of solving this problem, but it doesn’t seem that the proposed method is advancing research beyond this narrow setup.
- The real setup is pretty narrow, I wonder how much tuning went into the real setup given its small size and how easy it is to obtain good results with tuning.
- There is no mention or precise articulation / plan of how the techniques used here are transferable to the real world besides the analogy to how human infants learn from busyboards.
- Although busyboard is a more relevant env to the real world than Atari games, it is not necessarily that much more convincing in terms of relevance to general real world robotics. Maybe authors can improve that front.
- Future work discussion could benefit from a vision on how to expand the learnings of this work beyond the narrow setup of the busy board.

**Summary Of Recommendation:**

The science seems sound. The motivation of self-supervision with simplified toy-like setups with multiple abilities being tested (interaction, reasoning, planning) is good. While a new public benchmark is good as well, the issue I see with the paper is the proposed learning solution not being general enough to be of value beyond this narrow setup.

---

> ### Author Response · Authors · 2022-08-24
> **Author's Response to Reviewer yhwf**
>
> We thank the reviewer for detailed suggestions and feedback. We are glad to see the value of the BusyBoard environment is recognized! Here are the answers to your questions and reference to the revised sections in the paper.
>
> ### **Generality of the BusyBoard Environment**.
> > How the proposed approach is applicable beyond this setup?
>
> > Future work discussion could benefit from a vision on how to expand the learnings of this work beyond the narrow setup of the busy board.
>
> With the BusyBoard environment, we hope to provide a simple yet effective tool for evaluating the fundamental capabilities of embodied intelligent systems. We believe these capabilities are critical and relevant to many real-world applications beyond the scope of the proposed BusyBoard environments.
>
> To demonstrate the learned reasoning skill can be applied beyond the scope of the BusyBoard environment, we further test our reasoning model in two kitchen scenes from AI2THOR with realistic home appliances and inter-object functional relations (adapted from prior work IFR-EXPLORE). See updated pdf Sec. 8. In short, we found the reasoning module of BusyBot is able to infer the correct inter-object functional relations with no additional training on the two scenes.
>
> We also include additional discussion in Sec 5.1, detailing the assumptions made by the algorithm, and discussing the potential paths for future work to extend the proposed framework to handle more general settings and applications.
>
>
> ### **Generality of the BusyBot Method**
>
> > It seems unlikely that it would be useful for broad learning in the real world as is, a more unified model (as opposed to the 8 modules used here) or more end-to-end design.
>
> We do not agree. The number of modules in a system is not a good indicator of its generality. In fact, if we compare our method with an end-to-end learning method (e.g., PPO and BC in Table 3), we can see that while end-to-end learning could achieve good performance on the training tasks, its performance is significantly lower for unseen scenarios, indicating its poor generalization capability. In contrast, our modularized approach is able to generalize well to new scenarios by breaking down the problem.
>
> Moreover, the goal of designing BusyBoard is to comprehensively evaluate an embodied agent’s ability in manipulation, reasoning, and planning, instead of evaluating one specific skill. We believe these three capabilities are fundamental for many robot applications.  More explanation can be found in the response of [Generality of the BusyBoard Environment].
>
> ### **The real setup is pretty narrow**
>
> The real-world setup is a proof-of-concept evaluation of our methods' sim2real transfer capability. We agree it is limited, however, while we only tested on one real-world busyboard, we manually modified the underlying inter-object relations of the board by rewiring the objects to increase the diversity of test configurations.
>
> ### **Image difference reward in real-world scenario.**
> This reward implementation shows that when we use an “amplified effect”, the reward function can be simplified to a binary reward computed from image difference. A similar idea of (amplified effect) could be applied in the real-world scenario, however, a refined distance function (e.g., distance in feature space instead of pixel space) is needed to improve robustness.
>
> ### **Lacking references for GNN papers**
> We have added references for GNN [1, 2, 3, 4, 5] in the paper. Thank you!
>
> ### **Code and sim available.**
> Yes, we have uploaded the code in the updated zip file.
>
> ---
> [1] Santoro, Adam, et al. "A simple neural network module for relational reasoning." Advances in neural information processing systems 30 (2017).
>
> [2] Battaglia, Peter W., et al. "Relational inductive biases, deep learning, and graph networks." arXiv preprint arXiv:1806.01261 (2018).
>
> [3] Zečević, Matej, et al. "Relating Graph Neural Networks to Structural Causal Models." arXiv preprint arXiv:2109.04173 (2021).
>
> [4] Lin, Wanyu, Hao Lan, and Baochun Li. "Generative causal explanations for graph neural networks." International Conference on Machine Learning. PMLR, 2021.
>
> [5] Ke, Nan Rosemary, et al. "Learning neural causal models from unknown interventions." arXiv preprint arXiv:1910.01075 (2019).

---

> ### Author Response · Authors · 2022-08-27
> **Last Day of Discussion**
>
> Dear Reviewer,
>
> Today is the last day of the discussion phase. Since we haven't hear back from you yet, we’d like to reach out again to check if there are additional questions or concerns that we can address. Thank you!

---

### Official Review · Reviewer_gcRk · 2022-07-31

**Originality:** Fair
**Technical Quality:** Fair
**Clarity Of Presentation:** Fair
**Impact:** 2

**Recommendation:**

Weak Reject: I recommend rejecting the paper, but will not argue for my recommendation if the majority of other reviewers have a different opinion.

**Summary:**

The paper describes, on the one hand, a scenario for robot learning, planning, and acting inspired in a toy for cognitive development for children, and, on the other hand, a learning and planning framework tailored to this specific scenario to let a robot learn to execute a given task. The authors claim that one of the main contributions resides in the characteristics of the environment that allows a robot to learn cause-effect relations through interaction and to use these relations to select a sequence of action to bring the scenario from an initial configuration to a target (goal) one. They also propose a set of algorithms for learning and representing these cause-effects in a robotic application.



**Issues:**

Reduce the description of the scenario (e.g. Sec. 3) and use the gained space to elaborate on the scientific contributions.

The paper is missing a section where the terminology, problem definition, and basic tools are properly introduced. It will be helpful if the authors provide concrete examples of complete interact-learn-plan processes.

The reward is defined from differences in images. This will generate problems in real scenarios with changing light conditions and non-causal events taking place from external contingencies. How the authors are planning to address this problem? Would the proposed algorithms fail in these cases?

The explanation of the exploration strategy is unclear and insufficient.

Line 200. It is not clear is BusyBot is just the planning approach or the framework comprising all the learning algorithms (interaction policy, cause-effect relations, etc.).

Justify how the values of the parameters in the experimental set-up are defined (23 images, training test 32/9, etc. Why these arbitrary values?)

Lines 243 and 245. Typos.

Some references are incomplete (volume, issues, pages,…).

**Quality Of The Limitations Section:**

Limitations are not well addressed

**Reviewer Expertise:**

4: The reviewer is confident but not absolutely certain that the evaluation is correct

**Robotics Focus:**

Relevant but unlikely to deploy to hardware in near future

**Strengths And Weaknesses:**

The tasks used for validation of learning and planning approaches are too simple and can be solved by traditional approaches. The integration of sense-plan-act mechanisms with learning through interaction is not novel.  The perception mechanisms using vision and deep learning may entail some novelty but this is not developed in the paper. A clear presentation of the theoretical background and (theoretical) improvements achieved with respect to the state of the art is still missing.

There are no interdependencies between consecutive actions in a plan. The search problem is simple and straightforward.

One potentially interesting contribution of this work might be to let the robot learn the primitives to manipulate objects that requires small displacement. Unfortunately, these methods are not properly described and contrasted with respect to the state of the art. The experimental evaluation does not allow to fully assess the validity of the method. The comparisons are carried out only with ablated versions of their system, but not against the state of the art.

The contributions are difficult to distil from the paper. The authors condense a large number of algorithms without providing enough details and justifications. The paper in its current state gives the impression that the proposed methods are rather ad-hoc, with arguable transferability to other problems and scenarios.

The supplementary video is well-edited and clear.



**Summary Of Recommendation:**

I suggest the authors to avoid presenting the BusyBoard scenario as a contribution and to elaborate on methods that potentially entail scientific contributions with respect to the state of the art.

The description of Sec. 4.1 is incomplete and confusing. It is not clear for me how the action policy is defined/learned. I encourage the authors to better describe this method: What actions are considered in the action space? What is the state/object space? How the interaction policy is initialized? The authors mention that the robot-object interaction points are learned from depth images. How this takes place? Is there any previous knowledge used to guide the learning? How the per-pixel affordance is learned/defined? I think these could entail appealing contributions but unfortunately it is not possible to assess their validity with the current state of the paper.

The word "prediction" already entails future events. I suggest to avoid using the phrase "prediction of the future". I encourage the authors to explore the large literature in task and motion planning and learning of prediction models and use the same terminology.

The description of the limitation is insufficient. I recommend the authors to better discuss scalability, transferability, and robustness.

---

> ### Author Response · Authors · 2022-08-24
> **Author's Response to Reviewer gcRk**
>
> We thank the reviewer for detailed suggestions and feedback. Here are the answers to your questions and reference to the revised sections in the paper.
>
> ### **The complexity of the task is low and avoids presenting the BusyBoard scenario as a contribution.**
>
> We do not agree with this assessment. As recognized by all other reviewers,  we believe that the proposed BusyBoard environment is a valuable contribution to the community. As BusyBoard provides
>
> > “Complements typical toy domains .. by providing a relatively rich set of inter-object interactions” and “BusyBoard's procedural nature lends itself nicely to explorations of transfer learning and distribution shift.” (Reviewer 55uE)
>
> > “ is interesting because it allows exercising integrated skills for embodied agents. Interaction, reasoning, and planning fused into one environment.” and “The simplicity of the observations and action spaces could be seen as a plus, to focus on the reasoning and planning parts of the problem.” (Reviewer 9A6a)
>
> > “a self-contained diverse environment that is good for robotics research.” (Reviewer yhwf)
>
> Furthermore, we also provide additional results that demonstrate the skills learned in the BusyBoard environment can be applied to other applications such as home robotics. See Sec. 8 of the updated pdf.
>
> ### **Deep neural work is overkilled for relation learning.**
>
> We agree that **if the input is the canonical state of each object (e.g. binary state of light)**, the relation learning task can be solved by basic concept learning. However, a larger learning capacity can accommodate complex and high-dimensional input (e.g images), without the need for a hand-crafted state extractor, which makes the system easier to be designed, deployed, and generalized to novel scenarios.
>
> ### **The search problem is simple and straightforward.**
>
> While the planning task seems simple and straightforward, it requires the method to fully integrate the learned interaction and reasoning skills. In table 3, we compare with model-free RL algorithm (PPO) and behavior cloning method (BC) trained with oracle demonstration.
>
> The result shows that both PPO and BC methods tend to overfit on the training data and fail to generalize to boards with novel configurations or novel objects. The poor generalization result shows that the explicit representation of relations in our “simple” method is more effective to solve the planning task and generalize to scenes with novel configurations and objects.
>
> ### **Interaction part is not contrasted with respect to the state of the art.**
>
> Our interaction module is built on top of UMPNet [1], a state-of-the-art method for articulated object manipulation that relies on ground-truth object states for training. The major contribution is to leverage inter-object functional relations to amplify the visual feedback of small-displacement objects (e.g., switches) so that we can learn to infer action affordances from pure visual observations without the knowledge of ground truth object states.
>
> ### **Image difference reward in real-world scenario.**
>
> This reward implementation shows that when we use an “amplified effect”, the reward function can be simplified to a binary reward computed from image difference.  A similar idea of (amplified effect) could be applied in the real-world scenario, however, a refined distance function (e.g., distance in feature space instead of pixel space) is needed to improve robustness.
>
> ### **The description of Sec. 4.1 is incomplete and confusing.**
>
> The interaction policy outputs a sequence of 6D actions (parameterized as end effector position + direction). The interaction policy is initialized with random positions and directions, and we use the epsilon-greedy method to balance the exploration of random actions and exploitation of actions learned by the model. The model is trained in an online manner with two alternating phases: a data collection phase that stores interaction data into a replay buffer, and a training phase that randomly samples data points from the replay buffer for training.
>
> The training is guided by an exploration policy (Sec. 4.1 Exploration.), no other prior knowledge is used to guide the training. The details of the position and direction model and the training details are presented in the supplementary pdf. The position network is a standard UNet.

---

> > ### Author Response · Authors · 2022-08-24
> > **Author's Response to Reviewer gcRk**
> >
> > ### **The contributions are difficult to distill from the paper.**
> >
> > BusyBot represents a carefully designed initial attempt for the integrated task of Interact-Reason-and-Plan. While it is built on top of several existing methods, it has improved and integrated them in a unique way. We demonstrated its advantages over other alternative approaches through ablation studies. Here is a high-level summary:
> > 1.  The interaction module is built on-top-of UMPNet. However, leveraging amplified effects eliminates the need to obtain object state information from the simulation for reward computation.
> > 2. The reasoning module is adapted from V-CDN [2], however, our method extends the model to take as input raw features directly extracted from images instead of hand-crafted state representations (e.g., keypoints).
> > 3. Our planning module provides a new solution that effectively combines the inferred inter-object relationship and dynamics. This design allows efficient exploration, accurate planning, and generalization to novel configurations and objects, which have not been explored in prior works.
> >
> > ### **Typo and incomplete reference**
> > We have updated the reference and manuscript.
> >
> > ### **Usage of “prediction” and  “predict future state”.**
> >
> > In machine learning literature “predictions” is often used to generally refer to the output of a network, we use  “predict future state” to disambiguate these cases.
> >
> > ### **BusyBot in Line 200 is a planning approach or framework.**
> >
> > Here BusyBot represents the full framework comprising all learning modules.
> >
> > ### **Justify how the values of the parameters in the experimental set-up are defined**
> >
> > The hyper-parameters in our experiments are set up according to standard practices (e.g., setup used in V-CDN and other prior works) which follows an 80/20 split.  For example, within one interaction sequence that includes 30 images (29 interaction steps), we split it into relation extraction and future prediction segments: 23 (i.e., 29 * 80%) are used for relation extraction, and the rest 6 steps (i.e., 7 images) are used for future prediction. The same rule applied for object splits:  we have 41 switches in total and we randomly split them into training (32) and testing (9) sets based on the ratio of 80/20.
> >
> > ### **The description of the limitation is insufficient.**
> >
> > We have updated our limitation Sec 5.5 to include more discussions.
> >
> > ### **The explanation of the exploration strategy is unclear and insufficient.**
> >
> > We have described the exploration method in Sec. 4.1 and included additional explanations on how it works.
> >
> > ---
> > [1] Xu, Zhenjia, Zhanpeng He, and Shuran Song. "UMPNet: Universal manipulation policy network for articulated objects." arXiv preprint arXiv:2109.05668 (2021).
> >
> > [2] Li, Yunzhu, et al. "Causal discovery in physical systems from videos." Advances in Neural Information Processing Systems 33 (2020): 9180-9192.

---

> > > ### Comment · Reviewer_gcRk · 2022-08-26
> > > **The contributions are difficult to distill from the paper**
> > >
> > > "The contributions are difficult to distill from the paper."
> > >
> > > I appreciate the detailed enumeration of the contributions. These contributions should be highlighted already in the introduction and should be validated with experimental evidence.
> > >
> > > "BusyBot represents a carefully designed initial attempt for the integrated task of Interact-Reason-and-Plan."
> > >
> > > Regarding this comment and contribution #3, there are other works that integrate learning cause-effect through interaction, inferring changes in object relations with actions, and planning using the learned cause-effects to solve tasks having action interdependencies (a more complex planning problem than the one presented here). See, for example,
> > >
> > > Quack, B., Wörgötter, F., and Agostini, A. (2015). Simultaneously learning at different levels of abstraction. In 2015 IEEE/RSJ International Conference on Intelligent Robots and Systems (IROS) (pp. 4600-4607).
> > >
> > > Agostini, A., Torras, C., and Wörgötter, F. (2017). Efficient interactive decision-making framework for robotic applications. Artificial Intelligence, 247, 187-212.
> > >
> > > I suggest the authors to include these works in the state of the art, highlighting the differences of their integrated approach (interact-reason-plan) with respect to them.

---

> > ### Comment · Reviewer_gcRk · 2022-08-26
> > **BusyBoard scenario as a contribution**
> >
> > I appreciate the authors effort to provide clear responses to my concern.
> >
> > "The complexity of the task is low and avoids presenting the BusyBoard scenario as a contribution."
> > I do not argue against the utility of the scenario as a benchmark (as highlighted by the other reviewers). My concern is that using this scenario as a benchmark does not entail a significant scientific contribution. The contributions should be complemented by the methods proposed in the paper.
> >
> > I appreciate the additional experiment provided in Sec. 8. This also addresses my concern about transferability. One minor comment, I recommend the authors to place this and the previous section in the paper before the conclusions.
> >
> > "Deep neural work is overkilled for relation learning."
> > The answer to this concern is satisfactory.
> >
> > The paper still needs a major revision of the text and structure: Sec. 4.1, a section introducing basic elements, etc.

---

> ### Author Response · Authors · 2022-08-27
> **Second update and response**
>
> The authors would like to thank the reviewer again for all the detailed feedback! We have updated our manuscript to reflect these suggestions and attached the updated manuscript to this message. The second round of updates is labeled in brown.
>
> > The paper still needs a major revision of the text and structure: Sec. 4.1, a section introducing basic elements, etc.
>
> We have updated the manuscript with a better description of the interaction module, including an section introducing problem formulation.
>
> >Highlight contributions in the BusyBot algorithm
>
> We have updated the introduction to provide a detailed summary of contributions in the BusyBot algorithm. We also update the method and evaluation section to highlight these contributions.
>
> >  Compare with other works that tackle similar tasks.
>
> Thank you for the reference. We have added them into our related work section, with discussions on the differences from our methods. See the "Hierarchical Planning in Dynamic Environment" section.
>
> > I recommend the authors to place this and the previous section in the paper before the conclusions.
>
> We have updated the manuscript and will make sure to include this result in our final submission.

---

### Author Response · Authors · 2022-08-24
**Author’s response summary + updated manuscript + Code**

**Comment:**

We would like to thank the reviewers for their time and effort in helping us improve our work. We are glad to see the value of the proposed BusyBoard recognized. We have uploaded a revised manuscript (together with the original supplementary material for easy access). Below is a summarization of our response and major updates.

### **Generality beyond the BusyBoard environment.**
With the BusyBoard environment, we hope to provide a simple yet effective tool for evaluating the fundamental capabilities of embodied intelligent systems. We believe these capabilities are critical and relevant to many real-world applications beyond the scope of the proposed BusyBoard environments.

To demonstrate the learned skills can be applied beyond the BusyBoard environment, we further test our reasoning model in **two kitchen scenes from AI2THOR** with realistic home appliances and inter-object functional relations (adapted from prior work IFR-EXPLORE [1]). See updated pdf Sec. 8 for more discussion.   In short, we found the reasoning module of BusyBot is able to infer the correct inter-object functional relations with no additional training on both scenes.

### **Comparison to other methods**
To further validate our design decisions, we include additional baselines as suggested by reviewers:
- Baseline: we include an RL PPO baseline and a BC baseline in Table 3, which validates the importance of learning explicit representation of inter-object relations.
- RGB observation only baseline for manipulation: We include this baseline in Table 1, validating the choice of using depth input, which captures structure information and enables better generalization.

### **More complex many-to-one tasks**
We have included additional evaluation results on environments with many-to-one relations and discussed the potential ambiguity of such tasks. See updated pdf Sec. 7.

### **Novelty**

**Novelty of BusyBoard Environment**

As recognized by multiple reviewers,  we believe that the proposed BusyBoard environment is a valuable contribution to the community. As BusyBoard provides

> “Complements typical toy domains .. by providing a relatively rich set of inter-object interactions” and “BusyBoard's procedural nature lends itself nicely to explorations of transfer learning and distribution shift.” (Reviewer 55uE)

> “The environment is interesting because it allows exercising integrated skills'' and “The simplicity of the observations and action spaces could be seen as a plus, to focus on the reasoning and planning parts of the problem.” (Reviewer 9A6a)
> “a self-contained diverse environment that is good for robotics research.” (Reviewer yhwf)

**Novelty of BusyBot**

To the best of our knowledge, there is no existing work to solve the joint task of 1) inferring action affordances,  2) reasoning about inter-object functional relationships, and 3) performing goal-conditioned manipulation in a unified framework.  Therefore, BusyBot represents a carefully designed initial attempt for this task. While it is built on top of existing methods, it has improved and integrated them in unique ways. For example:
- The interaction module is built on top of UMPNet [2]. However, leveraging amplified effects eliminate the need to obtain object state information from the simulation for reward computation.
- The reasoning module is adapted from V-CDN [3], however, our method extends the model to take as input raw features directly extracted from images instead of hand-crafted state representations (e.g., keypoints).
- Our planning module provides a new solution that effectively combines the inferred inter-object relationship and dynamics. This design allows efficient exploration, accurate planning, and generalization to boards with novel configurations and objects, which have not been explored in prior works.

---
[1] Li, Qi, et al. "IFR-Explore: Learning Inter-object Functional Relationships in 3D Indoor Scenes." arXiv preprint arXiv:2112.05298 (2021).

[2] Xu, Zhenjia, Zhanpeng He, and Shuran Song. "UMPNet: Universal manipulation policy network for articulated objects." arXiv preprint arXiv:2109.05668 (2021).

[3] Li, Yunzhu, et al. "Causal discovery in physical systems from videos." Advances in Neural Information Processing Systems 33 (2020): 9180-9192.

**Zip File:**

/attachment/8e1ee550095c29c82bb810e7bfd2650c804604ad.zip

---

### Author Response · Authors · 2022-08-25
**Follow-up on the Responses**

Dear Reviewers,

We’d like to reach out again to check if there are additional questions or concerns about our rebuttal that we can address before the reviewer-author discussion period ends on August 28. Thanks again for taking the time to read our work and provide helpful feedback!

Paper Authors

---

### Meta-Review · Area_Chair_9rqr · 2022-08-06

**Recommendation:** Accept (Poster)
**Confidence:** 4

**Metareview:**

The paper proposes a method to learn cause-effect relationships on Busyboards in a self-supervised way.

The reviewers see strength in the paper of creating the Busyboard simulation environment as a benchmark and evaluating their approach on a real hardware setup.
However, the reviewers also raised many concerns about the generality and novelty as well as comparison to other methods.
I invite the authors to respond to the reviewers.

Post-rebuttal:
The authors have cleared many concerns by providing baseline evaluations and experiments on additional environments (2 envs from AI2THOR). I have read the revised version in detail and came to the following conclusion:
Although the reviews are still mixed, I believe that the paper adds value to the community, both by providing the Busyboard environments, as well as by creating a relatively simple algorithm for such interaction tasks, that shows good performance.

Comments: please rearrange the paper to fit the page limit and make sure the important aspects mentioned in the reviews make it to the main paper, whereas details are moved to the supplementary.
(Typo: L386: broads)